# Topical Tranexamic Acid Can Be Used Safely Even in High Risk Patients: Deep Vein Thrombosis Examination Using Routine Ultrasonography of 510 Patients

**DOI:** 10.3390/medicina58121750

**Published:** 2022-11-29

**Authors:** Yong Bum Joo, Young Mo Kim, Byung Kuk An, Cheol Won Lee, Soon Tae Kwon, Ju-Ho Song

**Affiliations:** 1Department of Orthopedic Surgery, Chungnam National University Hospital, Chungnam National University College of Medicine, Daejeon 35015, Republic of Korea; 2Department of Radiology, Chungnam National University Hospital, Chungnam National University College of Medicine, Daejeon 35015, Republic of Korea; 3Department of Orthopedic Surgery, Chungnam National University Sejong Hospital, Chungnam National University College of Medicine, Sejong 30099, Republic of Korea

**Keywords:** tranexamic acid, venous thromboembolism, total knee arthroplasty, transfusion

## Abstract

*Background and Objectives*: Previous studies regarding tranexamic acid (TXA) in total knee arthroplasty (TKA) investigated only symptomatic deep vein thrombosis (DVT), or did not include high risk patients. The incidence of DVT including both symptomatic and asymptomatic complications after applying topical TXA has not been evaluated using ultrasonography. *Materials and Methods*: The medical records of 510 patients who underwent primary unilateral TKA between July 2014 and December 2017 were retrospectively reviewed. Because TXA was routinely applied through the topical route, those who had a history of venous thromboembolism, myocardial infarction, or cerebral vascular occlusive disease, were not excluded. Regardless of symptom manifestation, DVT was examined at 1 week postoperatively for all patients using ultrasonography, and the postoperative transfusion rate was investigated. The study population was divided according to the use of topical TXA. After the two groups were matched based on the propensity scores, the incidence of DVT and the transfusion rate were compared between the groups. *Results:* Of the 510 patients, comprising 298 patients in the TXA group and 212 patients in the control group, DVT was noted in 22 (4.3%) patients. Two patients had DVT proximal to the popliteal vein. After propensity score matching (PSM), 168 patients were allocated to each group. In all, 11 patients in the TXA group and seven patients in the control group were diagnosed with DVT, which did not show a significant difference (*p* = 0.721). However, the two groups differ significantly in the transfusion rate (*p* < 0.001, 50.0% in the TXA group, 91.7% in the control group). *Conclusions:* The incidence of DVT, whether symptomatic or asymptomatic, was not affected by the use of topical TXA. The postoperative transfusion rate was reduced in the TXA group. Topical TXA could be applied safely even in patients who had been known to be at high risk.

## 1. Introduction

Total knee arthroplasty (TKA) leads to substantial blood loss and poses a risk of transfusion [1,2]. Because allogenic blood transfusion is associated with several complications, such as allergic reactions and metabolic imbalances [3,4], surgeons have made an effort to reduce postoperative blood loss. The use of tranexamic acid (TXA), a synthetic amino acid derivative of lysine which prevents fibrin degradation [5,6,7,8], has been proven efficacious in reducing blood loss after TKA, and thus the rate of allogenic transfusion [9,10,11].

A substantial body of research has endorsed the efficacy and safety of TXA [2,8,12,13]. However, previous studies often excluded patients with medical comorbidities that were associated with postoperative thromboembolism. There remains a concern that TXA may increase the risk of thromboembolic complications [14,15,16,17]. In particular, patients undergoing TKA are prone to deep vein thrombosis (DVT) because surgical trauma and tourniquet application accelerate local fibrinolytic activity [18]. The incidence of DVT after arthroplasty has been investigated based on manifested symptoms in most studies [19,20,21]; thus, the incidence would only indicate the tip of the iceberg [22,23]. 

The topical use of TXA has an equivalent effect in reducing the transfusion rate after TKA, compared to intravenous (IV) use. Recent guidelines published by the American Association of Hip and Knee Surgeon (AAHKS) stated that the strong supporting evidence for use of TXA in high risk patients was lacking [24]. At our institution, topical TXA was applied without restrictions in high risk patients, such as those who had history of venous thromboembolism, myocardial infarction, or cerebral vascular occlusive disease. This study aimed to determine whether the use of topical TXA affected the true incidence of DVT using ultrasonography, regardless of symptom manifestation. 

## 2. Materials and Methods

Medical records of 510 patients who underwent primary unilateral TKA between July 2014 and December 2017 were retrospectively reviewed after approval was obtained from our institutional review board. Because TXA was routinely applied through the topical route after September 2015, those who had history of venous thromboembolism, myocardial infarction, cerebral vascular occlusive disease, or cancer [25], were not excluded. The study population was divided according to the use of topical TXA.

### 2.1. Study Intervention

All TKAs were performed by two senior surgeons with the same surgical principle and prosthesis (Scorpio NRG, Stryker, Mahwah, NJ, USA). In the TXA group, 3.0 g of TXA in 100 mL of saline solution was applied directly into the knee joint cavity while suturing. An intra-articular drain was left for 48 h postoperatively. After the removal of drains, a range of motion exercise and walker-aided ambulation were encouraged. For DVT prophylaxis, intermittent pneumatic compression device was routinely used and a low molecular weight heparin was injected for 1 week after surgery [26,27]. There was no long-term anticoagulant therapy, regardless of the groups.

### 2.2. Study Design and Propensity Score Matching

DVT was examined using ultrasonography (Philips HD15, Bothwell, WA, USA) by two experienced radiologists. Lower extremity ultrasonography was performed at 1 week postoperatively for all patients, regardless of symptom manifestation. Proximal DVT was defined as the thrombosis that occurred proximal to the popliteal vein, and distal DVT was defined as the thrombosis of the anterior and tibial, peroneal, gastrocnemial, and soleal veins.

Postoperative transfusion records were reviewed, and the transfusion rate was compared according to the use of TXA. The necessity for transfusion was determined based on the guidelines by the National Institutes of Health Consensus Conference: hemoglobin level < 8.0 g/dL, or hemoglobin level < 10.0 g/dL with intolerable anemic symptoms or any anemia-related organ dysfunctions.

The two groups were matched at a 1:1 ratio based on propensity score [28]. The relevant variables were applied in a logistic regression analysis to calculate propensity scores. The incidence of DVT and the transfusion rate were compared between the groups after propensity score matching (PSM).

### 2.3. Statistical Analysis

The sample size of each group (168 patients) was confirmed by post hoc analysis to achieve a power of 96% to reject the null hypothesis with regard to the incidence of DVT, with a significance level of 0.05. Post hoc power analysis was performed using G*Power (Version 3.1.7, Franz Faul, Christian-Albrechts-Universitätzu Kiel, Kiel, Germany). The propensity score variables included age, sex, body mass index (BMI), American Society of Anesthesiologists (ASA) score, smoking, hypertension, diabetes mellitus, chronic kidney disease, arrhythmia, blood profiles (platelet count, prothrombin time (PT), and activated partial thromboplastin time (aPTT)), and medical comorbidities (cerebrovascular accident, myocardial infarction, other thromboembolism, and cancer). Propensity scores were matched using one-to-one nearest neighbor matching with no replacement and no caliper width. The matched patients were selected randomly to avoid a potential bias that came from an imbalance in the number of patients between the groups. After PSM, intergroup comparison of each variable was performed to confirm the validity of the matching. Categorical variables including the incidence of DVT and the transfusion rate were analyzed by Chi-square test when the expected value of the cell was 5 or more in at least 80% of the cells; otherwise, Fisher exact test was used. Continuous variables were analyzed by *t* test. All statical analyses were performed using the R software version 4.1.1 (R foundation for Statistical Computing, Vienna, Austria), with a *p* value < 0.05 considered statistically significant.

## 3. Results

A total of 510 patients with a mean age of 69.8 ± 7.6 years (range, 53–86 years) were followed up for 39.6 ± 23.5 months (range, 12–86 months). There were 298 patients in the TXA group and 212 patients in the control group (Table 1). Overall, DVT was noted in 22 (4.3%) patients, with two patients having DVT proximal to the popliteal vein. Computed tomography pulmonary angiography was performed in those patients, and none of them showed pulmonary embolism. No patients exhibited symptoms for DVT. The odds ratio of topical TXA was 1.03 (95% confidence interval 0.43–2.45).

After PSM, 168 patients were allocated to each group (Table 2). In all, 11 patients in the TXA group and seven patients in the control group were diagnosed with DVT. One patient in each group had DVT proximal to the popliteal vein. There was no significant difference in the incidence of DVT between the groups (*p* = 0.721). A total of 84 (50.0%) patients in the TXA group and 154 (91.7%) patients received allogenic transfusion, which did not show a significant difference between the groups (*p* < 0.001; Table 3).

## 4. Discussion

The primary finding of the present study was that topical TXA did not increase the incidence of DVT that was evaluated using ultrasonography. In this study, postoperative ultrasonography was performed in all patients to find both symptomatic and asymptomatic thrombosis, which could be the source of other serious complications such as pulmonary thromboembolism and cerebral infraction. The incidence of DVT represented the potential risk of TXA that could trigger systemic coagulation. This study proved that topical TXA reduced postoperative transfusion rate in TKA, which is consistent with previous studies [12,18,29].

It is surprising that concerns over the safety of TXA has not been fully addressed, given that TXA is a widely used modality to reduce the risk of transfusion after TKA. Based on recent guidelines by AAHKS, the recommendation for use of TXA in high risk patients is limited due to a lack of strong evidence [24]. There have been several randomized controlled trials (RCT) or level 1 studies investigating the efficacy and safety of TXA; however, those studies excluded high risk patients [13,14,15,29,30,31]. In their notable study, Whiting et al. performed a retrospective review of 1002 total joint arthroplasty patients with ASA score ≥ 3 to evaluate the outcomes of high risk patients who received TXA [19]. They found no differences in 30-day postoperative symptomatic thromboembolic events and postoperative transfusion rate. However, a concern regarding the potential procoagulant effect of TXA still remained because only symptomatic events were assessed. Besides, the indication of TXA was different among surgeons in their study, which would cause selection bias. Sabbag et al. also reviewed 1262 primary total hip or knee arthroplasty patients with a history of DVT [21]. There was no difference in the incidence of recurrent DVT between patients who received TXA and those who did not receive TXA. However, asymptomatic DVT and medical comorbidities could not be investigated in detail because the study was based on total joint registry data.

Another remarkable study by Jules-Elysee et al. compared local and systemic levels of thrombogenic markers and TXA between IV TXA group and topical TXA group [32]. They collected peripheral and wound blood samples, measuring levels of plasmin-anti-plasmin (PAP, a measure of fibrinolysis), prothrombin fragment 1.2 (PF1.2, a marker of thrombin generation), and TXA. The authors recommended a single dose of IV TXA because no major difference was found in the mechanism of action, coagulation, and fibrinolytic profile between topical TXA and a single dose of IV TXA. However, postoperative (1 and 4 h after tourniquet release) systemic PF1.2 levels were higher than intraoperative levels in both IV TXA group and topical TXA group. The procoagulant effect of TXA cannot be ruled out although TXA was not the only reason of increased PF1.2 levels. In the present study, the incidence of DVT was investigated as an indicator of thromboembolic complications by topical TXA.

This study proved the efficacy of topical TXA in reducing the transfusion rate after TKA. The AAHKS guidelines stated that a superior method of administration could not be identified [24]. The aforementioned study by Jules-Elysee et al. concluded that a single dose of IV TXA was preferrable to topical TXA because of its convenience without making a significant difference in coagulation profile, compared to the latter. However, the safety of IV TXA in high risk patients has not been strictly tested. Based on the present study, topical TXA could be considered in high risk patients undergoing TKA.

Several limitations should be noted. First, the retrospective nature of this study brought potential bias. The ideal design in examining the effects of TXA would be RCT. However, as in the previous RCT studies [18,29,32], including high risk patients in the study population would face an ethical issue. Although this study was based on retrospective data, PSM was applied to minimize the possible confounding effects. Second, this study showed relatively high transfusion rates, compared to other studies [19,29]. Because our hospital is the only tertiary referral center in the region, TKA patients often have multiple comorbidities that necessitate low threshold of transfusion. This can be the reason of the high transfusion rates. Third, DVT might have occurred after ultrasonographic examination that was performed at 1 week postoperatively. However, the delayed thrombosis can hardly be attributed to TXA, considering the short half-life of it.

## 5. Conclusions

The incidence of DVT, whether symptomatic or asymptomatic, was not affected by the use of topical TXA. Postoperative transfusion rate was reduced in the TXA group. Topical TXA could be applied safely even in patients who had been known to be at high risk.

## Figures and Tables

**Table 1 medicina-58-01750-t001:** Patient characteristics between the TXA and the control groups before PSM.

	Overall	Topical TXA		*p* Value
		TXA Group (*n* = 298)	Control Group (*n* = 212)	
Age, y	69.8 ± 7.6	70.1 ± 8.0	69.6 ± 7.3	0.527
Male/Female, n	74/436	44/254	30/182	0.899
BMI, kg/m^2^	26.3 ± 3.8	26.4 ± 4.0	26.1 ± 3.5	0.355
ASA score	2.2 ± 0.4	2.2 ± 0.4	2.2 ± 0.4	0.623
Smoking, n	35	15	20	0.074
Hypertension, n	344	199	145	0.703
Diabetes mellitus, n	119	55	64	0.002
Chronic kidney disease, n	33	19	14	0.926
Arrhythmia, n	23	13	10	0.856
Blood profiles				
Platelet count, 10^3^/μL	247.3 ± 73.7	248.4 ± 75.5	245.7 ± 71.1	0.683
PT, INR	1.0 ± 0.4	1.0 ± 0.6	1.0 ± 0.2	0.790
aPTT, sec	32.8 ± 4.8	32.6 ± 5.4	33.0 ± 2.4	0.416
Medical comorbidities				
Cerebrovascular accident, n	35	18	17	0.478
Myocardial infarction, n	48	27	21	0.760
Other thromboembolism, n	22	11	11	0.508
Cancer, n	14	9	5	0.787

TXA, tranexamic acid; PSM, propensity score matching; BMI, body mass index; ASA, American Society of Anesthesiologists; PT, prothrombin time; INR, international normalized ratio; aPTT, activated partial thromboplastin time.

**Table 2 medicina-58-01750-t002:** Patient characteristics between the TXA and the control groups after PSM.

	Overall	Topical TXA		*p* Value
		TXA Group (*n* = 168)	Control Group (*n* = 168)	
Age, y	69.7 ± 7.6	69.3 ± 7.2	70.0 ± 8.0	0.447
Male/Female, n	43/293	20/148	23/145	0.744
BMI, kg/m^2^	26.3 ± 3.7	26.2 ± 3.8	26.4 ± 3.5	0.529
ASA score	2.2 ± 0.4	2.2 ± 0.4	2.2 ± 0.4	0.547
Smoking, n	22	9	13	0.388
Hypertension, n	222	112	110	0.908
Diabetes mellitus, n	70	33	37	0.687
Chronic kidney disease, n	26	14	12	0.689
Arrhythmia, n	17	7	10	0.620
Blood profiles				
Platelet count, 10^3^/μL	248.0 ± 67.4	246.5 ± 65.0	249.4 ± 69.9	0.697
PT, INR	1.0 ± 0.5	1.0 ± 0.7	1.0 ± 0.3	0.954
aPTT, sec	32.8 ± 5.2	32.7 ± 6.2	32.9 ± 4.1	0.662
Medical comorbidities				
Cerebrovascular accident, n	26	11	15	0.541
Myocardial infarction, n	32	14	18	0.578
Other thromboembolism, n	18	10	8	0.809
Cancer, n	7	3	4	0.702

TXA, tranexamic acid; PSM, propensity score matching; BMI, body mass index; ASA, American Society of Anesthesiologists; PT, prothrombin time; INR, international normalized ratio; aPTT, activated partial thromboplastin time.

**Table 3 medicina-58-01750-t003:** Comparison of outcomes between the TXA group and the control group.

	TXA Group (*n* = 168)	Control Group (*n* = 168)	*p* Value
DVT			0.721
Distal to the popliteal vein	10	6	
Proximal to the popliteal vein	1	1	
Transfusion	84	154	<0.001

TXA, tranexamic acid; DVT, deep vein thrombosis.

## Data Availability

The manuscript has associated data, which will be deposited in repositories if necessary.

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
