# Peer review of "Topical Tranexamic Acid Can Be Used Safely Even in High Risk Patients: Deep Vein Thrombosis Examination Using Routine Ultrasonography of 510 Patients"

_medicina, 2022, doi:10.3390/medicina58121750_

Round 1

Reviewer 1 Report

The Authors show the manuscript entitled "Topical tranexamic acid can be used safely even in high risk patients: deep vein thrombosis examination using routine ultrasonography of 510 patient"

The topic is interesting and actual.

Some comments:

- Methods. What was the rationale on variables choice to include in propensity score analysis? The Authors didn't include only those resulted different between baseline characteristics.

- Methods. I suggest to move the propensity score variables in to statistical analysis paragraph.

- The Authors should further specify TXA use is not feasible in overall TKA patients. For example, another limitation should be in patients with history of cancer/active cancer underwent to TKA. The Authors should add this ref to discuss this point Canonico ME, et al. Venous Thromboembolism and Cancer: A Comprehensive Review from Pathophysiology to Novel Treatment. Biomolecules. 2022 Feb 4;12(2):259.

- Do the Authors can provide additional info on long-term follow-up of the enrolled patients?

Author Response

Some comments:

- Methods. What was the rationale on variables choice to include in propensity score analysis? The Authors didn't include only those resulted different between baseline characteristics.

We tried to investigate every available variable, including demographics and comorbidities that had been known to be associated with DVT. Although statistically significant (or borderline) differences were found only in diabetes mellitus and smoking in inter-group comparisons, propensity score matching was performed to avoid any possible confounding effects of factors other than TXA use.

- Methods. I suggest to move the propensity score variables in to statistical analysis paragraph.

The manuscript has been revised accordingly.

- The Authors should further specify TXA use is not feasible in overall TKA patients. For example, another limitation should be in patients with history of cancer/active cancer underwent to TKA. The Authors should add this ref to discuss this point Canonico ME, et al. Venous Thromboembolism and Cancer: A Comprehensive Review from Pathophysiology to Novel Treatment. Biomolecules. 2022 Feb 4;12(2):259.

As noted in the manuscript (Line 90–92), topical TXA was routinely applied and those who had history of venous thromboembolism, myocardial infarction, cerebral vascular occlusive disease, or cancer (newly included as suggested) were not excluded. The reference has been added.

- Do the Authors can provide additional info on long-term follow-up of the enrolled patients?

We agree that the long-term results would give more solid evidence for the safety of topical TXA. Unfortunately, the long-term data were not available because this study focused on the effect of topical TXA that was administered intraoperatively.

Reviewer 2 Report

This is a historical cohort study designed to estimate whether the use of topical Tranexamic acid in TKA is linked to an increase in Venous thromboembolism post surgery.  As all patients were screened post surgery, both asymptomatic and symptomatic VTE was recorded.  Propensity scoring was used to match both groups.  The study is clearly presented however I find the statistical analysis simplistic, and I am not convinced that the study was adequately powered.

English grammar in the article needs improvement.

I have the following comments.

1.Please define the primary outcome variable for the study

The sample size of each group (168 patients) was confirmed by post hoc analysis to 101 achieve a power of 96% to reject the null hypothesis, with a significance level of 0.05.

2. Please give more details of this calculation, what event rate was this based on and what was the non-inferiority margin?  

3. In the results section please give the odds ratio+/- confidence interval for VTE.

4. In the matched patients please give details of the any long term anticoagulant therapy (including aspirin), were there differences between the groups?

5. Please include a Kaplan Meier(cumulative incidence) curve for VTE.

Were there any pulmonary embolism recorded?

Please delineate asymptomatic and symptomatic VTE in the results.

No details of ethical approval are given.

What was the duration of followup, was it limited to 1 week?  Please discuss incidence of VTE post surgery over longer followup.  

Author Response

1.Please define the primary outcome variable for the study

The sample size of each group (168 patients) was confirmed by post hoc analysis to 101 achieve a power of 96% to reject the null hypothesis, with a significance level of 0.05.

The manuscript has been revised accordingly.

2. Please give more details of this calculation, what event rate was this based on and what was the non-inferiority margin?  

The calculation was based on the incidence of DVT. There was non-inferiority margin because it was a post hoc analysis and the effect size was 0.22.

3. In the results section please give the odds ratio+/- confidence interval for VTE.

The description has been added in the manuscript (Line 163–164).

4. In the matched patients please give details of the any long term anticoagulant therapy (including aspirin), were there differences between the groups?

During the study period, we did not perform any long-term anticoagulant therapy, regardless of the groups. This has been added in the manuscript (Line 102–103).

5. Please include a Kaplan Meier(cumulative incidence) curve for VTE.

Because exact time points at which DVTs occurred were not investigated and the incidence of DVT was assessed using ultrasonography at 1 week postoperatively, a Kaplan Meier curve could not be made.

Were there any pulmonary embolism recorded?

The description has been added in the manuscript (Line 161–162).

Please delineate asymptomatic and symptomatic VTE in the results.

The manuscript has been revised accordingly (Line 163).

No details of ethical approval are given.

The study was conducted in accordance with the Declaration of Helsinki, and approved by the Institutional Review Board of Chungnam National University Hospital (No. 2021-09-084, 2021.02.05.) (Line 249–251).

What was the duration of followup, was it limited to 1 week?  Please discuss incidence of VTE post surgery over longer followup

Because this study focused on the adverse effect of topical TXA that was applied intraoperatively, the long-term incidence of DVT was not investigated. This limitation was discussed in the manuscript (Line 228–230).